

# *Munidopsis* species (Crustacea: Decapoda: Munidopsidae) from carcass falls in Weijia Guyot, West Pacific, with recognition of a new species based on integrative taxonomy

Dong Dong[1,2,*], Peng Xu[3,*], Xin-Zheng Li[1,2,4,5] and Chunsheng Wang[3,6]

[1] Department of Marine Organism Taxonomy & Phylogeny, Institute of Oceanology, Chinese Academy of Sciences, Qingdao, People's Republic of China
[2] Center for Ocean Mega-Science, Chinese Academy of Sciences, Qingdao, People's Republic of China
[3] Key Laboratory of Marine Ecosystem and Biogeochemistry, State Oceanic Administration & Second Institute of Oceanography, Ministry of Natural Resources, Hangzhou, People's Republic of China
[4] University of Chinese Academy of Sciences, Beijing, People's Republic of China
[5] Laboratory for Marine Biology and Biotechnology, Pilot National Laboratory for Marine Science and Technology (Qingdao), Qingdao, People's Republic of China
[6] School of Oceanography, Shanghai Jiao Tong University, Shanghai, People's Republic of China
[*] These authors contributed equally to this work.

Corresponding authors
Dong Dong, dongd@qdio.ac.cn
Chunsheng Wang, wangsio@sio.org.cn

## ABSTRACT

Several squat lobster specimens of the genus *Munidopsis* were collected from an artificially placed carcass fall (cow bones) on Weijia Guyot in the western Pacific Ocean. Based on morphological comparisons and molecular analysis, three specimens were confirmed as juveniles of *M. albatrossae Pequegnat & Pequegnat, 1973*, which represents the first record of this species in the western Pacific. The other specimens collected are newly described as *Munidopsis spinifrons* sp. nov., which is distinguished from the closely related species in having a spinose rostrum and basal lateral eyespine on the eyestalk. The *M. albatrossae* from Weijia Guyot exhibited very low genetic distances when compared with a conspecific sample from Monterey Bay, USA, and the closely related species *M. aries* (*A. Milne Edwards, 1880*) from the northeastern Atlantic. A phylogenetic tree based on the mtCOI gene shows *M. spinifrons* sp. nov. as sister to *M. vrijenhoeki Jones & Macpherson, 2007* and *M. nitida* (*A. Milne Edwards, 1880*), although *M. vrijenhoeki* presents a complex relationship with other species in the clade. The systematic status of the new species and the closely related species are discussed.

## INTRODUCTION

The genus *Munidopsis Whiteaves, 1874* comprises more than 260 species of squat lobsters, distributed worldwide, and is one of the most diverse groups of deep-sea animals (*Baba et al., 2008*). *Munidopsis* species are adapted to a variety of abyssal habitats, such as seamounts, ocean plains, hydrothermal vents, cold seeps and whale falls. Whale falls or other types of
large organic falls are unique ecosystems creating an island-type habitat that can sustain a specifically adapted benthic community, potentially enduring for decades (*Smith et al., 1989*; *Baco & Smith, 2003*; *Smith & Baco, 2003*; *Pop Ristova et al., 2017*). The fauna in a carcass fall is often diverse and usually shows a high degree of endemism (*Baco & Smith, 2003*; *Smith & Baco, 2003*; *Amon et al., 2013*; *Amona et al., 2017*).

Known as opportunists, *Munidopsis* species have often been observed in whale-fall and wood-fall ecosystems (*Williams, Smith & Baco, 2000*; *Jones & Macpherson, 2007*; *Macpherson, Amon & Clark, 2014*; *Sumida et al., 2016*; *Amona et al., 2017*). Previous studies found that *Munidopsis* species associated with whale falls or wood falls had a heterogeneous diet and could be scavengers, predators, bacterivorous detritivores and/or even wood-eaters, and were present in the community in every developmental stage (*Janßen, Treude & Witte, 2000*; *Smith & Baco, 2003*; *Kemp et al., 2006*; *Macavoy et al., 2008*; *Hoyoux et al., 2012*). Although *Munidopsis* species constitute an important part of organic-fall fauna, there has been little taxonomic study of the group in this ecosystem compared with other habitats (*Jones & Macpherson, 2007*; *Macpherson, Amon & Clark, 2014*).

To study the fauna and community characters of a deep-sea carcass-fall ecosystem, artificially placed carcass falls (cow bones) were deployed at Weijia Guyot in the western Pacific, in 2016 and 2018, by the Second Institute of Oceanography, Ministry of Natural Resources of the People's Republic of China. Two landers containing cow bones mimicking whale falls were placed on the seamount at a depth of 1,427 m and 3,225 m, respectively. Several specimens of *Munidopsis* were collected among the fauna, together with many amphipod scavengers. After careful examination, three specimens were identified as juveniles of *M. albatrossae Pequegnat & Pequegnat, 1973*, a first record for this species in the West Pacific, and the other five specimens were found to be new to science. In addition to morphological analysis, we performed a barcoding analysis using sequences of the mtCOI gene to help the identification of specimens. Meanwhile, we selected a group of *Munidopsis* species morphologically similar to our specimens for phylogenetic analysis, most of which have mesial eyespines, relatively short P1 and triangular rostrum, and are generally associated with chemosynthetic environments (*Baba & De Saint Laurent, 1992*; *Jones & Macpherson, 2007*; *Cubelio et al., 2007*; *Coykendall, Nizinski & Morrison, 2017*; *Rodríguez-Flores, Macpherson & Machordom, 2018*). In the present article, we combined the morphological with molecular methods in an attempt to clarify the systematic status of the present species from Weijia Guyot and assess their phylogenetic relationships with other congeners.

## MATERIALS & METHODS

### Sample collection and morphological examination

Information on the *Munidopsis* species collected at Weijia Guyot and the molecular data used in the present study are listed in Table 1. The carcass fall experiments were conducted using deep-sea landers including cow bones deployed on the seabed. The squat lobsters were collected when the landers were retrieved one year after the deployment. All specimens collected were found on the bones or within the lander boxes. After being photographed,

**Table 1** Species, vouchers ID, Genbank accession numbers and references in this study.

| Species | Vouchers/sample codes | GenBank accession numbers | References |
|---|---|---|---|
| *Munidopsis spinifrons* sp. nov. | SRSIO18100001 | MN397915 | Present study, holotype |
| *Munidopsis spinifrons* sp. nov. | SRSIO18100002 | MN397916 | Present study, paratype |
| *Munidopsis spinifrons* sp. nov. | SRSIO18100002 | MN397917 | Present study, paratype |
| *Munidopsis spinifrons* sp. nov. | SRSIO18100002 | MN397918 | Present study, paratype |
| *Munidopsis spinifrons* sp. nov. | SRSIO18100002 | MN397919 | Present study, paratype |
| *Munidopsis albatrossae* | SRSIO1709000X | MN397920 | Present study |
| *Munidopsis lauensis* | M159_19 | MN397921 | Present study |
| *Munidopsis nitida* | M202 | MN397922 | Present study |
| *Munidopsis nitida* | M203 | MN397923 | Present study |
| *Munidopsis albatros sae* | USNM 1101472 | DQ677692 | *Jones & Macpherson (2007)* |
| *Munidopsis antonii* | | DQ677686 | *Jones & Macpherson (2007)* |
| *Munidopsis aries* | | DQ677691 | *Jones & Macpherson (2007)* |
| *Munidopsis barbarae* | MNHN-IU-2014-13822 | MG979479 | *Rodríguez-Flores, Macpherson & Machordom (2018)* |
| *Munidopsis bermudezi* | J2282_01 | KX016541 | *Coykendall, Nizinski & Morrison (2017)* |
| *Munidopsis bracteosa* | | DQ677684 | *Jones & Macpherson (2007)* |
| *Munidopsis cascadia* | USNM 1100637 | DQ677694 | *Jones & Macpherson (2007)* |
| *Munidopsis corniculata* | MNHN-IU-2013-19128 | MG979481 | *Rodríguez-Flores, Macpherson & Machordom (2018)* |
| *Munidopsis exuta* | | DQ677690 | *Jones & Macpherson (2007)* |
| *Munidopsis kensmithi* | SIO C10973 | DQ677709 | *Jones & Macpherson (2007)* |
| *Munidopsis livida* | J2282-02 | KX016546 | *Coykendall, Nizinski & Morrison (2017)* |
| *Munidopsis myojinensis* | NSMT-Cr16877 | EF143603 | *Cubelio et al. (2007)* |
| *Munidopsis recta* | SIO C10969 | DQ677702 | *Jones & Macpherson (2007)* |
| *Munidopsis scotti* | D751_5 | KY581548 | *Goffredi et al. (2017)* |
| *Munidopsis similis* | 4179_01 | KX016549 | *Coykendall, Nizinski & Morrison (2017)* |
| *Munidopsis verrucosus* | SIO C10881 | DQ677710 | *Jones & Macpherson (2007)* |
| *Munidopsis vrijenhoeki* | | DQ677676 | *Jones & Macpherson (2007)* |
| *Munidopsis vrijenhoeki* | | DQ677675 | *Jones & Macpherson (2007)* |
| *Shinkaia crosnieri* | | KR003157 | *Shen et al. (2016)* |

the specimens were preserved in 80% ethanol. The size of the specimen is given as the postorbital carapace length (PCL), which refers to the carapace length excluding rostrum. All specimens collected in this study were deposited in the Sample Repository of Second Institute of Oceanography (SRSIO), Ministry of Natural Resources, Hangzhou, China. Field experiments were approved by the China Ocean Mineral Resources R & D Association (Cruise 41 and 58).

*Munidopsis lauensis* Baba & de Saint Laurent, 1992 was used as the comparative material for the phylogenetic analysis. The specimen was collected from a hydrothermal vent in Manus Basin, Bismarck Sea (3°42.25′S, 151°52.66′E), at 1,714 m depth, in 19 June 2015. Morphological and molecular data for specimens of *M. nitida* (*A. Milne Edwards, 1880*) used in the study were made available by Paula Rodríguez-Flores, Museo Nacional de Ciencias Naturales (MNCN-CSIC) and Centre d'Estudis Avançats de Blanes (CEAB-CSIC),

Spain. Those specimens were collected off Papua New Guinea (03°31′S, 148°03′E), at a depth range of 780–855 m, in 23 April 2014.

## Molecular data and analysis

Total genomic DNA was extracted from muscle tissue using QIAamp DNA Mini Kit (QIAGEN, Hilden, Germany), following manufacturer instructions. Extracted DNA was eluted in double-distilled $H_2O$ (dd$H_2O$). Partial sequences of the COI gene were amplified via polymerase chain reaction (PCR). Reactions were carried out in a 30-μl volume containing: 15 μl Premix Taq (TaKaRa Taq$^{TM}$ Version 2.0 plus dye; TaKaRa, Kusatsu, Japan), 1.2 μl each of forward and reverse primers (10 mM), respectively, 1.6 μl DNA template, and 11 μl dd$H_2O$. For *M. albatrossae*, the primer pair gala_COIF and gala_COIR was used to amplify a fragment of 568 base pairs (bp) of the COI gene following the original procedure (*Jones & Macpherson, 2007*). For *M. spinifrons* sp. nov. specimens, we designed a new reverse primer LCOgala (5′- ATCATAAAGACATTGGAACTTTATA - 3′) paired with the universal forward primer HCO2198 (*Folmer et al., 1994*) to obtain a fragment of ca. 660 bp of the COI gene in the following thermal profile: initial denaturation at 95 °C for 5 min; 40 cycles of 95 °C for 50 s; 49 °C for 50 s; 72 °C for 50 s; and a final extension at 72 °C for 10 min. We also amplified a fragment (672 bp) of the COI gene of *M. lauensis* for phylogenetic analysis using the universal primers HCO2198 and LCO1490 (*Folmer et al., 1994*) following the thermal profile described above. PCR products were purified using a QIAquick Gel Extraction Kit (QIAGEN, Hilden, Germany), and bidirectionally sequenced using the same primers with an ABI 3,730 ×l Analyzer (Applied Biosystems, Foster City, CA, USA).

Sequences were checked by the sequence peak height and then assembled based on the contigs using the DNASTAR Lasergene software package (DNASTAR, Inc., Madison, WI, USA). The sequences acquired during this study were uploaded to NCBI GenBank (Table 1). We also downloaded several COI sequences of Munidopsidae species from GenBank (Table 1) for the phylogenetic analysis. Most of the chosen species, which are morphologically similar to either one of the two species presently studied, belong to the *Orophorhynchus* (*A. Milne Edwards, 1880*) group and associated with chemosynthetic environments (*Ahyong, Andreakis & Taylor, 2011*). Two morphologically distinct species, *M. barbarae* (*Boone, 1927*) and *M. corniculata Rodríguez-Flores, Macpherson & Machordom, 2018*, were chosen for comparative analysis, and *Shinkaia crosnieri Baba & Williams, 1998* was selected as the outgroup in the phylogenetic study.

The sequences were aligned using the software package MEGA 6.06 (*Tamura et al., 2013*). The average genetic distances within and between species were estimated according to the Kimura 2-parameter (*Kimura, 1980*) model in MEGA 6.06 (*Tamura et al., 2013*). The most appropriate nucleotide base substitution model for the alignment data, which is HKY+I+G, was determined by MrModeltest v2 (*Nylander, 2004*). The maximum likelihoods (ML) for phylogenetic analyses were assembled in PhyML 3.1 (*Guindon & Gascuel, 2003*) with 1,000 replicates. A Bayesian inference (BI) tree was constructed using MrBayes 3.2.6 (*Huelsenbeck & Ronquist, 2001*). Markov chains were run for 10,000,000 generations, sampled every 100 generations; the first 25% trees were discarded as burn-in, after which remaining trees

were used to construct the 50% majority-rule consensus tree and to estimate posterior probabilities.

## Zoobank registration

The electronic version of this article in portable document format will represent a published work according to the International Commission on Zoological Nomenclature (ICZN), and hence the new names contained in the electronic version are effectively published under that Code from the electronic edition alone. This published work and the nomenclatural acts it contains have been registered in ZooBank, the online registration system for the ICZN. The ZooBank Life Science Identifiers (LSIDs) can be resolved and the associated information viewed through any standard web browser by appending the LSID to the prefix http://zoobank.org/. The LSID for this publication is: urn:lsid:zoobank.org:pub:C1CC52FD-6113-4A53-91C3-8D80E713D255. The online version of this work is archived and available from the following digital repositories: PeerJ, PubMed Central, and CLOCKSS.

# RESULTS

## Taxonomy

Family Munidopsidae *Ortmann, 1898*
Genus *Munidopsis Whiteaves, 1874*
*Munidopsis albatrossae Pequegnat & Pequegnat, 1973*
(Figs. 1, 2 and 3A)

*Munidopsis* sp. *Wolff, 1961*: 148, fig. 16.
*Munidopsis albatrossae Pequegnat & Pequegnat, 1973*: 163, Figs. 1, 2 (type locality: Eastern Pacific South of Madalena Bay, Baja California).—*Baba, 2005*: 284.—*Jones & Macpherson, 2007*: 480, Fig. 2A.—García Raso et al., 2008: 1282, Fig. 2.
*Munidopsis aries Ambler, 1980*: 17.—*Wicksten, 1989*: 315 (not *M. aries A. Milne Edwards, 1880*).

**Materials examined.** SRSIO1709000X, 1 male (PCL 6.29 mm), 1 female (PCL 6.51 mm), 1 sex indet. (PCL 4.02 mm). Carcass fall experimental field site, Weijia Guyot, West Pacific. R/V *Haiyang* 6, stn. MCMX1605, 12°43.0149′N, 156°27.2057′E, 3,225 m, 30 September 2017.

**Diagnosis.** Carapace (Fig. 1A) (excluding rostrum) as long as broad. Frontal margins oblique, with blunt outer orbital angle above antennal peduncle. Anterolateral corners blunt, followed by notch at lateral end of anterior cervical groove and another low process; anterior branchial margins rugose, slightly convex, each with blunt anterior tooth; posterior branchial margins converging posteriorly. Dorsal surface with numerous rugae; gastric region elevated, with pair of low epigastric process. Rostrum broadly triangular, 1.3 times longer than broad, 0.6 times of remaining carapace length; lateral margins straight, weakly serrated distally; dorsal surface and ventral surface (Fig. 1B) each with median,

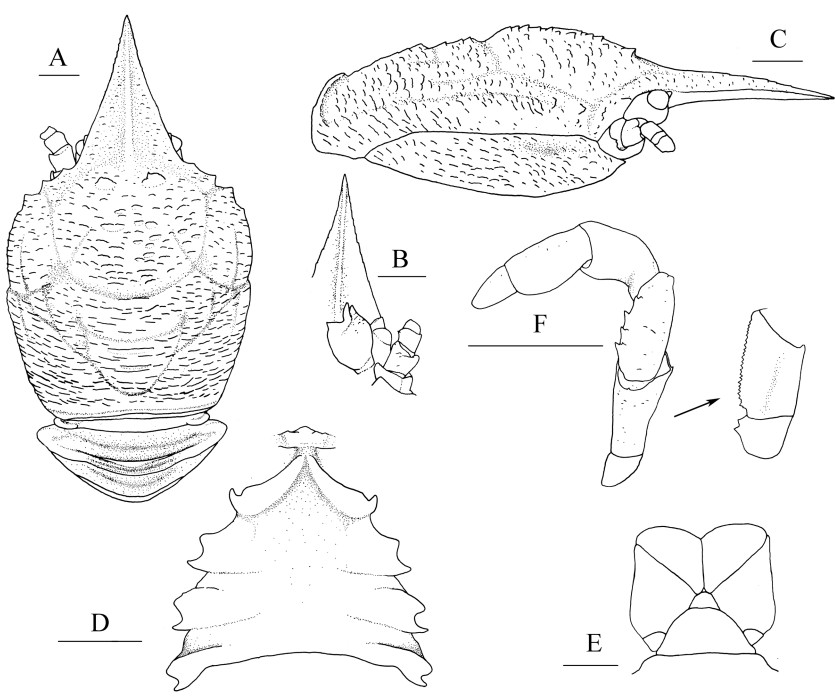

**Figure 1** *Munidopsis albatrossae Pequegnat & Pequegnat, 1973*, **SRSIO1709000X, male.** (A) Carapace and abdominal tergites 1–3, dorsal view; (B) rostrum, eye, left antennule and antenna, ventral view; (C) carapace and right pterygostomian, lateral view; (D) sternal plastron, ventral view; (E) telson, dorsal view; (F) left third maxilliped and ischium crista dentata, ventral view. Scales equal 1.0 mm.

longitudinal ridge. Pterygostomial flaps (Fig. 1C) with oblique rugae on lateral surface; anteriorly acute. Sternal plastron (Fig. 1D) as long as broad; sternite 4 moderately elongated anteriorly, ventral surface depressed. Abdominal tergites unarmed; tergites 2 and 3 each with two transverse ridges on dorsal surface, anterior ridges more elevated. Telson (Fig. 1E) composed of 8 plates. Eyestalks short, unmovable, mesial eyespines present but reduced; cornea small. Basal article of antennular peduncle (Fig. 1B) with 2 anterior spines. Antennal peduncle (Fig. 1B) overreaching eyestalks. Merus of third maxilliped (Fig. 1F) armed with 3 or 4 small spines on flexor margin. Pereopod 1 (P1, chelipeds) (Fig. 2A) subequal; palms with rows of spines on lateral and mesial margins, dorsal surfaces with scattered spines; fingers with opposable margins distally spooned; fixed fingers with denticulate carina on distolateral margin. Pereopods 2–4 slender (P2–4) (Figs. 2B–2E); meri spinulose on extensor and flexor margins; carpi each with 2 longitudinal, spinulose carinae on extensor surface; propodi each with 2 carinae on extensor surface and 1 pair of distal spines on flexor margin; dactyli (Fig. 2E) approximately 0.7 propodi length; flexor margin straight, with 7 elevated teeth, each bearing small corneous spines. P2 slightly overreaching distal end of P1. P1 with epipods.

**Coloration.** In preserved condition, grey white.

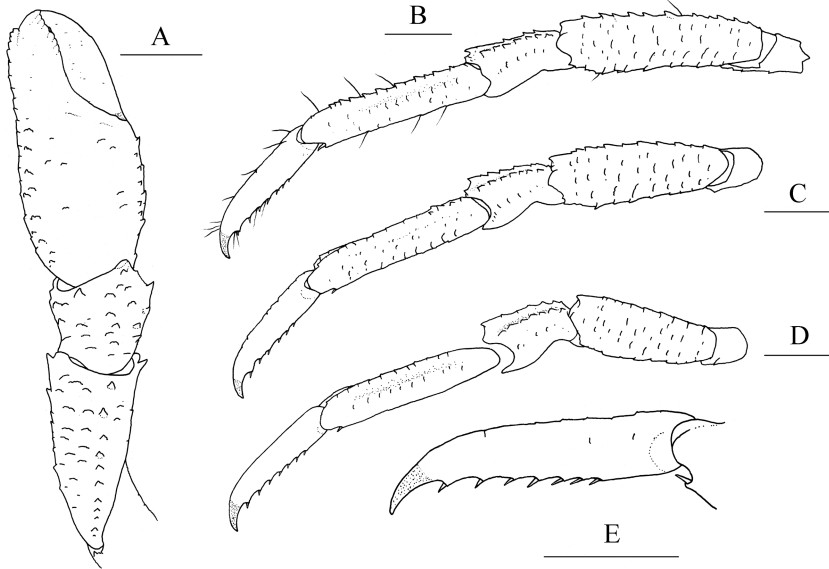

**Figure 2** *Munidopsis albatrossae Pequegnat & Pequegnat, 1973*, **SRSIO1709000X, male.** (A) left cheliped (P1), dorsal view; (B) left pereopod 2 (P2, with setae), lateral view; (C) left pereopod 3 (P3), lateral view; (D) left pereopod 4 (P4), lateral view; (E) dactylus of left P2, lateral view. Scales equal 1.0 mm.

**Distribution.** East Pacific: south of Madalena Bay, Baja California, Costa Rica, East Pacific Rise, and Monterey Bay (California). Antarctic waters: Bellingshausen Sea. West Pacific: Weijia Guyot. Depth 1,920–3,680 m.

**Habitats.** The specimens described here were collected from cow bones in an artificial carcass fall, deployed at 3,225 m at Weijia Guyot. This species was previously found on a whale fall in Monterey Bay (*Jones & Macpherson, 2007*), and on soft sea bottom in the Bellingshausen Sea (*García Raso, García Muñoz & Manjón-Cabeza, 2008*).

**Remarks.** The specimens from Weijia Guyot show a few differences from the holotype and the specimen from Monterey Bay. In the present specimens, the mesial eyespines are blunt and the anterior branchial margins are faintly serrated, whereas in the holotype, the mesial eyespines are prominent and the anterior branchial margins have numerous small spines (*Pequegnat & Pequegnat, 1973*). The present specimens are all juveniles, with the PCL not longer than 7 mm; in contrast, the specimens described from the East Pacific exceed 70 mm PCL. Therefore, the slight morphological differences can be considered intraspecific variations due to size.

### *Munidopsis spinifrons* sp. nov.

urn:lsid:zoobank.org:act:B1A381D7-7BF1-4C37-A15F-33ACF009C833

(Figs. 3B, 4 and 5)

*Munidopsis vrijenhoeki Jones & Macpherson, 2007*: 496 (part, small leg fragments?).

**Materials examined.** SRSIO18100001, holotype, 1 male (PCL 24.89 mm); SRSIO18100002, paratypes, 4 females (PCL 15.79–20.45 mm). Carcass fall experimental field site, Weijia

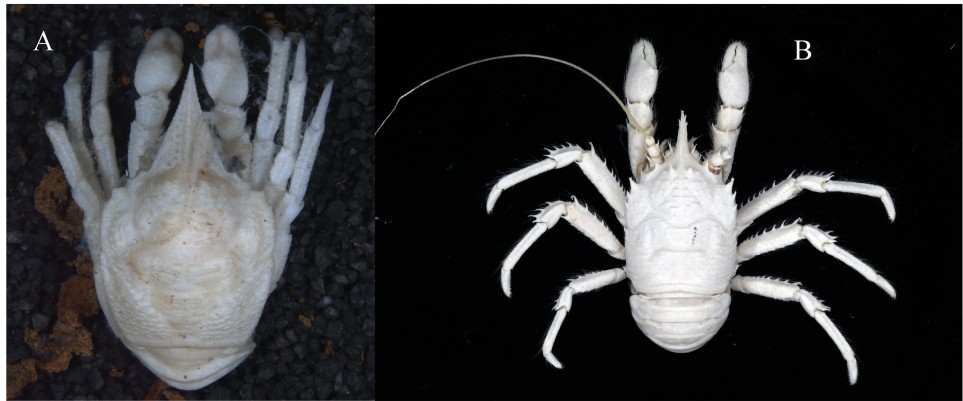

**Figure 3** **Photos of specimens examined, preserved in ethanol.** (A) *Munidopsis albatrossae Pequegnat & Pequegnat, 1973*, SRSIO1709000X, male; (B) *Munidopsis spinifrons* sp. nov., SRSIO18100001, holotype.

Guyot, West Pacific. R/V *Dayang 1*, stn. DY48-II-MX1802, 12°56.96′N, 156°57.25′E, 1427.5 m, 25 October 2018.

**Description.** Carapace (Fig. 4A) (excluding rostrum) distinctly longer than broad. Frontal margins oblique, antennal spines well developed. Lateral margins approximately parallel, bearing short, sparse setae. Anterolateral spines relatively short. Anterior branchial margin with 3 or 4 spines; anteriormost spine strongest; posterior two spines usually rudimental. Posterior branchial margin rugose, with distinct spine at base of posterior cervical groove. Posterior margin unarmed, slightly concave. Dorsal surface covered with transverse, interrupted ridges, bearing long setae. Gastric region elevated, with 2 strong epigastric spines (followed with 2 tiny spines in holotype). Cervical groove distinct. Cardiac region with distinct transverse uninterrupted ridge. Rostrum spiniform (Fig. 4B), 0.4 times as long as remaining carapace length, 1.2 times broader than long (base at level of antennal spine base); dorsal surface evenly and longitudinally carinate; distal 0.3 length of lateral margins strongly upturned, bearing 1 or 2 small but distinct spines. Pterygostomial (Fig. 4C) flaps with oblique rugae on lateral surface.

Sternal plastron (Fig. 4D) slightly longer than broad, widening posteriorly. Sternite 3 broader than anterior margin of sternite 4, divided into two parts by median longitudinal groove; anterior margins with median notch. Sternite 4 narrowly elongated with longitudinal groove in anterior part; posterior part broad, surface with short scales, posterior surface depressed. Sternites 5–7 each with elevated, transverse ridges, bearing simple setae.

Abdominal tergites smooth and unarmed; tergites 2–4 each with 2 transverse ridges bearing stiff setae anteriorly, posterior ridge relatively short.

Telson (Fig. 4E) composed of 10 distinct plates.

Eyestalk (Fig. 4F) hardly movable. Cornea oval, globular, broader than long. Ocular peduncle short, nearly invisible in dorsal view, broader than cornea; mesial eyespine prominent, anterolaterally directed, reaching to distal 0.6 of rostrum; lateral eyespine

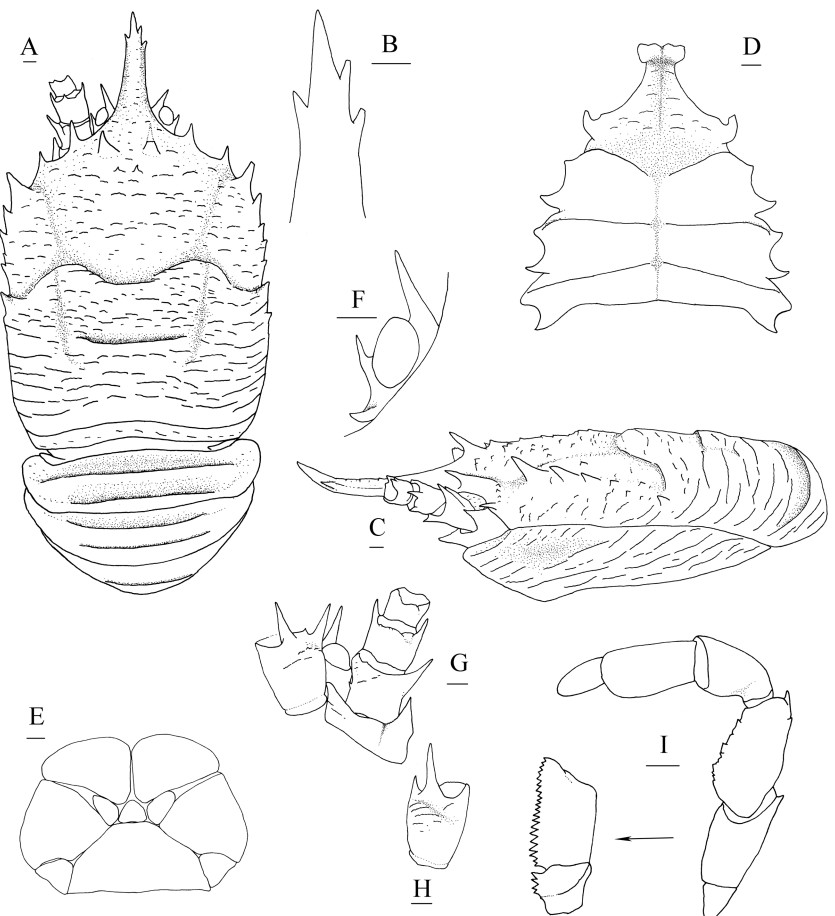

**Figure 4** *Munidopsis spinifrons* **sp. nov., SRSIO18100001, holotype.** (A) Carapace and abdominal ter-gites 1–3, dorsal view; (B) distal part of rostrum, dorsal view; (C) carapace and left pterygostomian, lateral view; (D) sternal plastron, ventral view; (E) telson, dorsal view; (F) left eyestalk, dorsal view; (G) left an-tennule and antenna, ventral view; (H) right antennule, ventral view; (I) left third maxilliped and ischium crista dentata, ventral view. Scales equal 1.0 mm.

relatively short, closely adjacent to cornea and followed with distinct spine on base of peduncle (usually covered by carapace in dorsal view).

Antennular peduncle (Figs. 4G, 4H) with basal article longer than broad; distal margin bearing strong ventrolateral spine and dorsolateral spine (rarely bearing another minute intermedian spine); lateral face slightly inflated, covered with short rugae; mesial margin straight.

Antennal peduncle (Fig. 4G) reaching to half length of rostrum, bearing setae on lateral and mesial margins. Article 1 immovable, with strong distomesial and distolateral spines. Article 2 armed with strong distolateral spine and small mesial spine at midlength. Article 3 subrectangular, with strong distomesial and distolateral spines, and minute dorsodistal spine. Article 4 short and unarmed.

Third maxilliped (Fig. 4I) slender. Ischium approximately as long as merus length, disto-extensor corner acute; crista dentata well-developed, extending onto basis. Merus

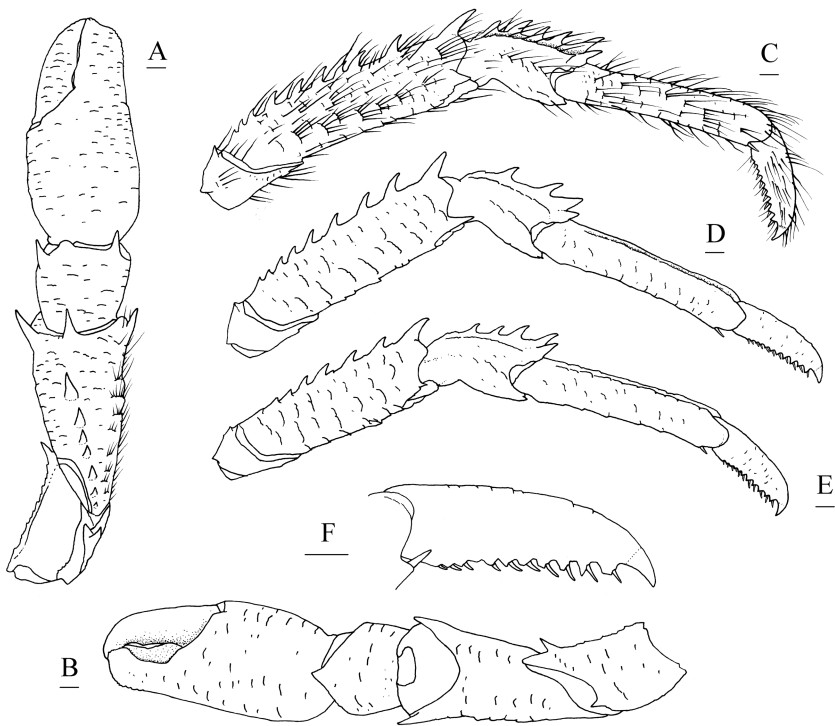

**Figure 5** *Munidopsis spinifrons* **sp. nov. SRSIO18100001, holotype.** (A) Right cheliped (P1), dorsal view (setae only shown on lateral margin of merus); (B) right cheliped (P1), ventral view; (C) right pereopod 2 (P2, with setae), lateral view; (D) right pereopod 3 (P3), lateral view; (E) right pereopod 4 (P4), lateral view; (F) dactylus of right P2, lateral view. Scales equal 1.0 mm.

subrectangular, extensor margin with distinct distal spine followed by small tubercle; flexor margin irregularly denticulate. Carpus unnamed. Propodus with distoflexor margin convex. Dactylus short. Dactylus flexor margin, propodus distoflexor margin, and carpus dorsal distoflexor margin densely covered in long setae.

Pereopod 1 (P1, chelipeds) (Figs. 5A, 5B) subequal, 1.4 times PCL, densely covered in long and stiff setae on rugae and base of spines on surface and margins. Ischium short, approximately 0.7 merus length, distal margin with distinct dorsolateral spine and small ventrolateral spine; ventrodistal margin anteriorly produced, with strong subterminal spine. Merus approximately 0.4 PCL, subtriangular in cross-section, with short rugea on surfaces; dorsal surface armed with longitudinal row of spines (strongest on distal margin, successively decreasing in size); dorsodistal margin with another strong spine on mesial side; ventrodistal margin with strong mesial and lateral spines. Carpus less than half merus length, dorsomesial margin with strong subdistal spine and small median spine (disappearing on right P1 of holotype); dorsolateral margin with strong distal spine; ventrodistal margin produced into triangular lobe. Chela relatively compressed, approximately 1.3 merus length (including fixed finger), twice as longer as broad; palm unarmed. Fingers 0.8 palm length, opposable margins distally spooned and crenulated; occlusal margins sinuous, with low, triangular tooth proximally on fixed finger, and broad,

low tooth medially on movable finger; distolateral margin of fixed finger with indistinct denticulate carina.

Pereopods 2–4 (P2–4, ambulatory legs 1–3) (Figs. 5C–5F) setose, bearing long, stiff setae on margins and surfaces of each segment. P2 approximately 1.8 times PCL, overreaching distal end of cheliped. Meri somewhat compressed; P2 merus approximately 0.7 times PCL (P3 merus 0.9 P2 merus length; P4 merus 0.8 P2 merus length), 4.3 times as long as broad (P3 3.8 times, P4 3.3 times); extensor margin armed with row of spines, distal-most spine prominent; flexor margin rugose, with strong distal spine. Carpi each with 2 longitudinal ridges on extensor surface; lateral carina rugose, armed with small distal spine (P2, sometimes absent) or unarmed (P3 and P4); mesial carina armed with row of 3–6 spines, distal spine laterally situated and subequal (or smaller) in size to penultimate spine; flexor margin armed with small but acute distal spine. Propodi subcylindrical, P2 propodus 0.8 merus length (P2–4 propodi subequal in length); extensor surface nearly flat, with 2 longitudinal carinae; flexor margin rugose, with pair of distal corneous spines. Dactyli (Fig. 5F) 0.4–0.5 propodi length; extensor margin rugose; flexor margin straight, with 11 movable corneous spines (increasing in size distally) each based on triangular tooth.

P1 with epipod.

**Distribution.** Known only from the type locality, Weijia Guyot, West Pacific; 1,427.5 m.

**Coloration.** In fresh condition, body entirely white, cornea light orange.

**Habitats.** The species is currently recorded only from the artificially placed deep-sea carcass fall (cow bones) at Weijia Guyot, western Pacific.

**Etymology.** Latin words "*spini-*" means spinose or spiny, and "*frons*" means the rostrum. The new specific name refers to the special character that discriminates it from the closely related species.

**Remarks.** The new species is morphologically similar to *M. nitida* in having a narrow rostrum, parallel carapace lateral margins, pair of strong epigastric spines, mesial and lateral eyespines, broad cornea, unarmed abdominal segments, anterior branchial margin with 2–4 spines, P1 shorter than P2, and epipod present on P1. *Munidopsis spinifrons* sp. nov. can be readily distinguished from *M. nitida* in having small distal spines on the lateral margins of the rostrum, and a basal lateral eyespine; in *M. nitida*, the lateral margins of the rostrum are entire (*Baba, 2005*) and the eyestalk bears only a distal lateral eyespine. The latter character has not been mentioned in previous literature, but according to the illustration in *Baba (2005*: Fig. 72e), *M. nitida* lacks such a basal lateral eyespine. This was also supported by examination of specimens from New Guinea (P Rodríguez-Flores, pers. comm., 2019). The character of the spinose rostrum can be observed on all five of the present specimens, regardless of the size and sex, although the spines are weak on the smallest specimen; therefore we accept it as a consistent and reliable interspecific character. The new species resembles *M. exuta Macpherson & Segonzac, 2005* in having a narrow rostrum with small lateral marginal spines and a pair of strong epigastric spines. *Munidopsis spinifrons* sp. nov. differs from *M. exuta* in having antennal spines, 10 telson plates, lateral eyespines, and denticulate carina on the P1 fixed finger.

COI sequence data (see below) show that the new species is closely related to *M. vrijenhoeki*. However, these two species differ morphologically. Besides the spinose

rostrum, the new species has a narrow rostrum, pair of strong epigastric spines, 10 telson plates, large cornea, anterior branchial margins with 2 or 3 spines, 3rd maxillipeds meri with irregular denticles on the flexor margins, P1 with epipods, and P2–4 meri each with a row of spines only on the extensor margin. In contrast, *M. vrijenhoeki* has a broad rostrum, pair of small epigastric spines, 8 telson plates, small cornea, branchial margins nearly unarmed, 3rd maxillipeds meri with well-developed spines on the flexor margin, P1 without epipod, and P2–4 meri with a row of spines on the flexor margin. The genetic relationships are discussed in the Discussion section.

### Molecular data analysis

Kimura's two-parameter pairwise genetic distances between *M. albatrossae* from the Weijia Guyot and a specimen from the Monterey Bay (*Jones & Macpherson, 2007*) was 0.4%, suggesting that the specimens are the same species. The genetic distances between *M. aries* from the northeastern Atlantic (*Jones & Macpherson, 2007*) and *M. albatrossae* from both the Weijia Guyot and Monterey Bay were 1.8% and 1.4%, respectively, indicating a close relationship between these two species.

Kimura's two-parameter pairwise genetic distance between *M. spinifrons* sp. nov. and *M. nitida* was 4%, indicating clear genetic divergence. However, *M. spinifrons* sp. nov., including all five sequences, showed no significant genetic distance compared with *M. vrijenhoeki* Mvri2 (DQ677675), yet displayed high genetic distance compared with *M. vrijenhoeki* Mvri3 (DQ677676), at 1.7%.

The combined phylogenetic trees (Fig. 6) reconstructed from both the ML and BI analyses are generally congruent. In the combined trees, *M. albatrossae* from both the west and east Pacific cluster together (BP [maximum likelihood bootstrap percentage] = 91), although the Bayesian posterior probability (PP) are modest. Meanwhile, *M. albatrossae* and *M. aries* form a highly supported monophyletic clade A (BP = 100, PP = 1.00) suggesting the close relationship of these two species.

In the phylogenetic tree (Fig. 6), *M. spinifrons* sp. nov., *M. vrijenhoeki* (both Mvri2 and Mvri3), and *M. nitida* form highly supported clade B (BP = 94, PP = 1.00), indicating their close relationship. The five specimens of the new species cluster together into a subclade, although the bootstrap value is modest (BP = 65); nevertheless, *M. vrijenhoeki* Mvri3 and *M. nitida* form a strongly supported subclade (BP = 98, PP = 1.00), illustrating that they are more genetically related than the rest of the species (individuals) within clade B. *Munidopsis lauensis* and *M. myojinensis Cubelio et al., 2007*, together with clade B, compose a large clade C with high Bayesian support (PP = 0.99).

## DISCUSSION

The COI gene is considered much conserved in the genus *Munidopsis*, following studies between populations and among sibling species (*Jones & Macpherson, 2007*; *Thaler et al., 2014*; *Coykendall, Nizinski & Morrison, 2017*). The smallest nucleotide divergences for mtCOI between species of *Munidopsis* from the EP Rise are 1.6%–1.9% (*Jones & Macpherson, 2007*). The genetic distance (Kimura's two-parameter pairwise), however, observed between *M. albatrossae* (from Monterey Bay) and *M. aries* (northeastern Atlantic)

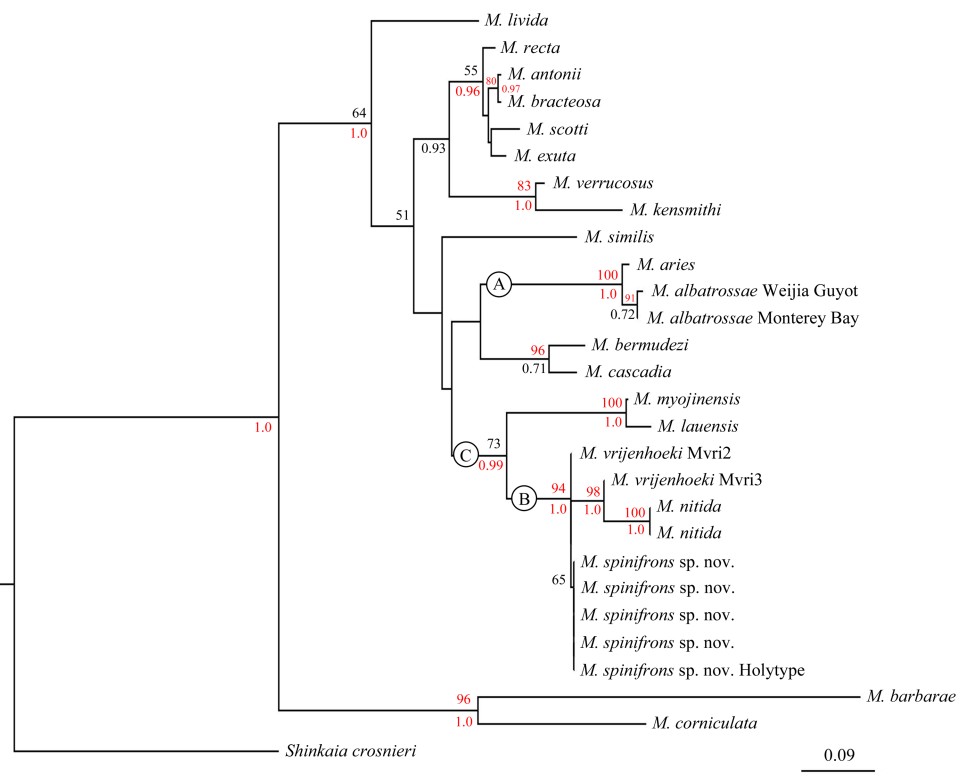

**Figure 6** **Phylogenetic tree obtained by the Maximum likelihood analysis based on the COI gene sequences.** Maximum likelihood bootstrap scores (BP, above) and Bayesian posterior probabilities (PP, below or right) are indicated adjacent to each node. Values of BP ≥ 75% and PP ≥ 0.95 are marked red. Only values of BP ≥ 50% and PP ≥ 0.5 are shown.

was 1.4%, which is smaller than that reported by *Jones & Macpherson (2007)* (2.8%), probably because data from different specimens were used. Similarly, *M. albatrossae* from the Weijia Guyot displayed very low genetic distance compared with specimens from the East Pacific, although their geographic distance is vast. *Harino et al. (2005)* mentioned the capture of *M. albatrossae* at the Nankai Through, off the south coast of Japan, but that record was not confirmed by taxonomic examination. Therefore, the current report is considered the first definite record of *M. albatrossae* distributed in the West Pacific. *García Raso, García Muñoz & Manjón-Cabeza (2008)* also reported the occurrence of this species in Antarctic waters (west of Peter I Island). Together, these findings suggest that *M. albatrossae* has a wide distribution range in the Pacific Ocean. Among the sampling localities reported in the present and previous studies, the Weijia Guyot and Monterey Bay are the only two places where the specimens were observed in a whale-fall or carcass-fall environment.

The type material of *M. vrijenhoeki* comprised of three specimens: the holotype and the small leg fragments of two other specimens (*Jones & Macpherson, 2007*); three COI sequences were published in GenBank based on these materials (DQ677674–DQ677676, representing Mvri1–Mvri3). The sequence of DQ677674 (Mvri1) was assigned to the

holotype, but annotated as including two individuals. However, as compared with other molecular data, the Mvri1 sequence was too short (211 bp, identical with part of Mvri2) to be used for phylogenetic analysis in the present study. The genetic distance observed between *M. spinifrons* sp. nov. and Mvri2 specimen was 0, but was significantly higher (1.7%) between *M. spinifrons* sp. nov. and Mvri3. Moreover, the combined phylogenetic tree confirms that Mvri3 is more closely related to *M. nitida* than to Mvri2 and *M. spinifrons* sp. nov., with high support. These results suggest that the type material of *M. vrijenhoeki* represents at least 2 valid species. Consequently, we think that one or two of the leg fragments attributed to *M. vrijenhoeki* might instead belong to the present new species, and that the Mvri3 sequence might be derived from the holotype. Still, the morphological differences were sufficiently distinct to distinguish our specimens as representing a valid species separate from *M. vrijenhoeki*. Nonetheless, additional samples accompanied by molecular examination involving multi-genes are needed to verify their actual systematic relationship.

The phylogenetic relationships among species within *Munidopsis* genus has been fully discussed (*Jones & Macpherson, 2007*; *Ahyong, Andreakis & Taylor, 2011*; *Coykendall, Nizinski & Morrison, 2017*). According to the phylogenetic tree in the present study (Fig. 6), the species (or specimens) from the West Pacific (except *M. albatrossae* in Weijia Guyot) are all clustered together in clade C, while species from the East Pacific and Atlantic are scattered in other clades, suggesting a level of genetic divergence between fauna from different geographic regions. However, some West-Pacific species (or specimens) are morphologically more similar to congeners from the East Pacific and Atlantic Ocean. For example, *M. spinifrons* and *M. nitida* from the West Pacific resemble the East-Pacific species, *M. bracteosa Jones & Macpherson, 2007* and *M. scotti Jones & Macpherson, 2007*, in having strong spines on the anterior branchial margins; on the contrary, *M. lauensis* and *M. vrijenhoeki* from the West Pacific lack such spines. The broad rostrum and small cornea also link *M. vrijenhoeki* to *M. aries* and *M. bermudezi Chace, 1939* in Atlantic Ocean. The current result supports the idea that there is no correlation between morphological and genetic divergences (at least based on COI) for squat lobsters (*Jones & Macpherson, 2007*). Since the COI gene of *Munidopsis* species is much conserved and indirectly correlated to the morphological differentiations, more barcoding genes are needed to be explored for the species identity, and multi-genes conjoint analysis is necessary to reveal the phylogenetic relationship among species from different geographic areas.

Juveniles of squat lobsters have seldom been described, either from a natural or artificial carcass fall or wood fall. The *M. albatrossae* collected from Weijia Guyot were all juveniles, with the longest being only 6.51 mm PCL; in contrast, specimens of this species from other localities reach 87 mm PCL (*Pequegnat & Pequegnat, 1973*). *Jones & Macpherson (2007)* also examined a juvenile collected from a whale fall. *Hoyoux et al. (2012)* found adults of a *Munidopsis* species within mesh boxes (containing woody baits), with body sizes larger than the mesh size, meaning that the species would have entered the boxes as larvae. The available evidence supports the view that large organic falls may attract *Munidopsis* larvae by acting as a nursery area, but where their growth into other life stages may be sustained. Carcass falls are considered as ''stepping stones'' for species endemic in chemosynthetic

environment to disperse over large distances (*Smith et al., 1989*; *Smith & Baco, 2003*; *Amon et al., 2013*). Accordingly, this unique habitat would function likewise for squat lobsters, which are widespread and opportunistic in this ecosystem.

## CONCLUSION

The new species *M. spinifrons* sp. nov. was established based on morphological and molecular studies. It is morphologically different with the closely related species in having spinose rostrum and basal lateral eyespine on the eyestalk. The new species had a very low COI genetic distance with *M. vrijenhoeki*, but further molecular analysis showed that the type material of *M. vrijenhoeki* contained at least two valid species, one of which might belong to the present new species. More sampling effort and multi-gene analysis are needed in the future to verify their actual systematic relationship. The discovery of juvenile individuals of *M. albatrossae* in this study supports the view that large organic falls in the deep sea may act as nursery area in some squat lobsters' life history, which is crucial for their dispersal over large distances.

## ACKNOWLEDGEMENTS

We are extremely grateful to Dr Paula Carolina Rodríguez-Flores and Prof. Enrique Macpherson for providing us with valuable morphological and molecular data on *M. nitida*. We extend sincere thanks to associate professor Dongsheng Zhang for help with the sample collection.

### Funding

This work was financially supported by the National Key R&D Program of China, (No. 2018YFC0310800); the China Ocean Mineral Resources R & D Association (COMRA) Special Foundation (No. DY135-E2-2-03, DY135-E2-2-06 and DY135-E2-3-04); the Senior User Project of RV KEXUE (No. KEXUE2018G25) and the National Natural Science Foundation of China (NSFC) (No. 31572229 and 41606155). The funders had no role in study design, data collection and analysis, decision to publish, or preparation of the manuscript.

### Grant Disclosures

The following grant information was disclosed by the authors:
National Key R&D Program of China: 2018YFC0310800.
China Ocean Mineral Resources R & D Association (COMRA) Special Foundation: DY135-E2-2-03, DY135-E2-2-06, DY135-E2-3-04.
Senior User Project of RV KEXUE: KEXUE2018G25.
National Natural Science Foundation of China (NSFC): 31572229, 41606155.

### Competing Interests

The authors declare there are no competing interests.

## Author Contributions

- Dong Dong conceived and designed the experiments, performed the experiments, analyzed the data, prepared figures and/or tables, authored or reviewed drafts of the paper, approved the final draft.
- Peng Xu conceived and designed the experiments, contributed reagents/materials/analysis tools, authored or reviewed drafts of the paper, approved the final draft.
- Xin-Zheng Li conceived and designed the experiments, authored or reviewed drafts of the paper, approved the final draft.
- Chunsheng Wang conceived and designed the experiments, contributed reagents/-materials/analysis tools, authored or reviewed drafts of the paper, approved the final draft.

## Field Study Permissions

The following information was supplied relating to field study approvals (i.e., approving body and any reference numbers):

Field experiments were approved by the China Ocean Mineral Resources R & D Association.

## DNA Deposition

The following information was supplied regarding the deposition of DNA sequences:

The COI sequences of the new species are available at GenBank: MN397915 to MN397919.

## Data Availability

The specimens described in this study are deposited in Sample Repository of Second Institute of Oceanography (SRSIO), Ministry of Natural Resources, Hangzhou, China. Voucher ID for *M. albatrossae* is SRSIO1709000X; vouchers ID for the holotype and paratypes of *M. spinifrons* are SRSIO18100001 and SRSIO18100002, respectively.

COI barcoding data is available at NCBI GenBank: MN397915 to MN397923.

## New Species Registration

The following information was supplied regarding the registration of a newly described species:

Publication LSID: urn:lsid:zoobank.org:pub:C1CC52FD-6113-4A53-91C3-8D80E713D255

*Munidopsis spinifrons* sp. nov. LSID: urn:lsid:zoobank.org:act:B1A381D7-7BF1-4C37-A15F-33ACF009C833

## Supplemental Information

Supplemental information for this article can be found online at http://dx.doi.org/10.7717/peerj.8089#supplemental-information.

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
