# Peer review of "Munidopsis species (Crustacea: Decapoda: Munidopsidae) from carcass falls in Weijia Guyot, West Pacific, with recognition of a new species based on integrative taxonomy"

_PeerJ, doi:10.7717/peerj.8089_

## Round 0.1 · original submission · Minor Revisions

Both reviewers have suggested minor revisions. Please consider them and submit a revised version of your manuscript.

·

Basic reporting

Straight forward paper, the authors present two species from an interesting environment and provide valid argument that one of them is new, using an integrative approach. well done.

Experimental design

I included a question as to what was actually placed on the ocean floor. You say bones, but I was curious to learn more. Can you direct the reader to a voyage report perhaps?

Validity of the findings

The authors present good illustrations and discussions of the species they discovered. I did suggest to delete the comment for the new species habitat, that it is suspected to also live on hydrothermal vents since there is currently little evidence to support this.
It might be useful to include a table of genetic distances in addition to a tree, but you do present a selection of these in the discussion.

Additional comments

I look forward to seeing it in print.

·

Basic reporting

The manuscript describes a new species of squat lobster of the genus Munidopsis and the occurrence of another rare species in the carcass fall of the western Pacific. The text is correct and the illustrations are adequate. The references, as well as the comparisons with other species is adequate. I have no problemas to accept the ms for publication after a few suggestions.

Experimental design

No comment

Validity of the findings

The findings are adequate. My minor comments are the following:
1. Line 45. Can you add a reference? e.g. Baba et al., 2008
2. Line 75. " In the present paper"
3. Line 93. Add affiliation of this colleague.
4. Line 176. Eye movable or fixed?
5. Line 217. Frontal margin oblique?
6. Line 273. Carpi. Dorsal or external surface?
7. Line 277. P4 propodus is slightly shorter than P2 and P3.
8. Line 283. Include depth.
9. Lines 305 and 310. and also different number of plates in the telson.
10. Discussion. Have you got any explanation of these low differences among species?

Additional comments

see above

---

## Round 0.2 · accepted · Accept

Corrections are OK so the manuscript is acceptable as it is.